# Quantitative analysis of lateral root development with time-lapse imaging and deep neural network

Yuta Uemura and Hironaka Tsukagoshi

Faculty of Agriculture, Meijo University, Nagoya, Japan

## Insights

deep neural network; lateral root development; lateral root primordium; time-lapse imaging; very-long chain fatty acid.

**Corresponding author:**
Hironaka Tsukagoshi;
Email: thiro@meijo-u.ac.jp

## Abstract

During lateral root (LR) development, morphological alteration of the developing single LR primordium occurs continuously. Precise observation of this continuous alteration is important for understanding the mechanism involved in single LR development. Recently, we reported that very long-chain fatty acids are important signalling molecules that regulate LR development. In the study, we developed an efficient method to quantify the transition of single LR developmental stages using time-lapse imaging followed by a deep neural network (DNN) analysis. In this 'insight' paper, we discuss our DNN method and the importance of time-lapse imaging in studies on plant development. Integrating DNN analysis and imaging is a powerful technique for the quantification of the timing of the transition of organ morphology; it can become an important method to elucidate spatiotemporal molecular mechanisms in plant development.

Elucidation of the molecular mechanisms through which root architecture is controlled is an important topic in plant development. Plant roots consist of primary root and lateral root (LR). For both root types, maintaining a balance between cellular proliferation and differentiation is key for adequate growth. The differentiation of LRs from the pericycle cells starting from lateral root primordia (LRP) formation can be considered a model case of novel tissue development. The primary root forms a series of cellular lineages from the root tip to the base; the cells in the developing LRP undergo dynamic changes in identity, at least until the LRP matures (Torres-Martínez et al., 2019). For such events, it is vital to control cellular function by regulating the function of phytohormones and their transcriptional in response to them (Banda et al., 2019; Fukaki & Tasaka, 2009). Although extensive genetic and molecular analyses have revealed the molecular mechanisms of root architecture development, root development is a continuous process; therefore, spatiotemporal analysis must be performed.

Time-lapse imaging is a powerful tool used to reveal continuous root growth regulations because it enables the observation of spatiotemporal changes such as molecular signalling, organelle behaviour and organ development. To understand the regulation of target tissue development by genes, it is necessary to link quantitative phenotypic information with gene expression through time-series analysis. Therefore, a method for quantitatively analysing phenotypic changes and gene expression in the same time series with a high temporal resolution by observing gene expression and phenotypes in living plants is required.

In *Arabidopsis thaliana*, fluorescence time-lapse imaging of GCaMP, a fluorescent protein-based cytosolic $[Ca^{2+}]$ sensor, revealed momentary and dynamic $Ca^{2+}$ signalling, which propagates to a leaf different from the one that accepts the signal at approximately 1 mm/s (Toyota et al., 2018). Microscopic live tracking of organelle behaviour involving the process of root cap cell detachment, together with cell organelles, has revealed that the intracellular position of the nucleus changes during root cap detachment (Goh et al., 2022). In the case of organ development, confocal microscopy time-lapse imaging has provided direct evidence of the division patterns of stem cells and the proximal meristem in root tips (Campilho et al., 2006). Furthermore, the

cell division mechanism of LRP has been investigated by continuously observing the number of cells at each developmental stage of LRP, revealing differences in LRP development in individual plants (Campilho et al., 2006; Lucas et al., 2013). It is critical to analyse and interpret continuous images captured via time-lapse imaging to elucidate seamless plant development. However, the number of images obtained from time-lapse imaging tends to be extremely high, and accurately discriminating and quantifying the growth of plants from such big data requires automation of image analysis. In this regard, analysis methods that apply machine learning are effective and indispensable. Digital image processing, including deep neural network (DNN) analysis, is an efficient and powerful technique for automatic image analysis. DNN analysis is superior to conventional digital image processing approaches in terms of computational programming simplicity. Conventional image recognition must quantify organ morphology through digital image processing, such as colour thresholding, feature extraction, statistics and algorithms. PlantCV provides a useful Python package for computational image processing that focuses on plant research (Fahlgren et al., 2015; Gehan et al., 2017). To date, 181 digital image analysis tools, such as BRAT, a root length measurement tool, and GiA Roots, a root system architecture characterisation tool, have been developed; however, there are no DNN-based image processing tools with high contrast and background noise-reduction features (Galkovskyi et al., 2012; Lobet, 2017; Lobet et al., 2013; Slovak et al., 2014). None of the DNN-based image-processing methods comprises multiple algorithms; however, the DNN model is composed of neural network layers, and a combination of these layers determines the accuracy and application target. A DNN allows us to develop an image-recognition model by training an image dataset with high accuracy without complicated computer programming.

Phenotypic analysis of time-lapse images requires the quantification of the developmental stage, cell or tissue size, length, and population size. The automatic recognition of developmental stages from image data is a typical task in DNN applications. Various DNN classification models have been developed, such as Resnet50 and Xception, that can be used for stage classification without the need for complicated analytical pipelines (Chollet, 2017; He et al., 2016). Additionally, cell size, cell length and population size estimation can be used to perform DNN image segmentation and processing. DNN segmentation models, such as U-Net and X-Net, have been proposed (Fujii et al., 2021; Ronneberger et al., 2015). Each cell tissue size in a developed callus has been quantified through DNN image segmentation and image processing, and the DNN enables the quantification of different cell sizes (Ikeuchi et al., 2022). PlantSeg has also been used for DNN segmentation and for cell length and volume quantification from Z-stack 3D images of roots and leaves captured using confocal laser microscopy (Graeff et al., 2021; Wolny et al., 2020). These studies demonstrate that DNN models can be applied to other studies and obtain robust, accurate, and even efficient results for the quantification of plant development.

Recently, we reported that very long-chain fatty acids (VLCFAs) are involved in LR development through the regulation of the expression of the transcription factor MYB93 (Uemura et al., 2023). In addition to MYB93, ATML1, which has a START domain (a lipid-binding region), reportedly binds to very long-chain ceramides and plays a key role in epidermal differentiation (Abe et al., 2003; Lu et al., 1996; Sessions et al., 1999). *pas2* mutants, which lack the function of VLCFA synthase, reportedly show abnormal cell proliferation in the shoot apical meristem

(Nagata et al., 2021; Nobusawa et al., 2013). These results indicate that VLCFA synthesis and signalling are required for plant development. Additionally, the 3-ketoacyl-CoA synthase 1 (*KCS1*) gene encoding one of the key VLCFA synthesis enzyme is characteristically expressed in the LRP and LRP-peripheral cells, suggesting the involvement of VLCFAs in LRP development. Supporting these findings, several studies have shown that lower VLCFA levels affect LR development (Shang et al., 2016; Trinh et al., 2019). We identified a transcription factor, *MYB93*, through RNA sequencing of a *kcs1-5*. The VLCFA levels decreased in *kcs1-5* mutants, indicating that MYB93 is a novel transcription factor whose expression responds to VLCFA levels. *MYB93* expression shows a specific response to fatty acid carbon chain length, with no response at C18 but a response to C20–C24 VLCFAs (Uemura et al., 2023). Moreover, genetic analysis has revealed that MYB93 is involved in the late stages of LR development by regulating the expression of several cell wall remodelling genes, such as expansins (Uemura et al., 2023).

In an analysis of LRP development under the regulation of VLCFAs and MYB93, we developed a machine learning-based method for quantifying LRP development. LRP development has conventionally been assessed by inducing LRP development and analysing the distribution of developmental stages by observing the LRP at several time points. In general, the gravistimulation assay has been used widely to count LRP stages (Lee et al., 2015; Péret et al., 2012). In this assay, gravistimulated roots are fixed at several time points, and the LRP developmental stages are counted under a microscope. This analysis provides information on LRP development-stage distribution in bulk samples. Therefore, a single LRP development cannot be traced, and there is a lack of information on the transition time among different LRP stages. We deduced that the combination of time-lapse imaging and DNN analysis would enable to measure the developmental transitions of a single LRP. Furthermore, conducting multi-time point experiments with different genotypes or chemical treatments increases the sample size significantly even within a single experiment, placing a substantial burden on researchers. Therefore, we developed a machine learning-based method for quantifying LRP development, as published in 'A very long chain fatty acid responsive transcription factor, MYB93, regulates LR development in Arabidopsis' (Uemura et al., 2023). Here, we describe the rationale, development and implications of the DNN approach for LR studies, as well as explore numerous other biological questions. We performed LRP time-lapse imaging using VLCFA-related mutants, such as *kcs1-5,* and image analysis using the DNN. For LRP time-lapse imaging, we developed a system that could capture time-lapse images under a microscope while the plants are growing on the medium and monitored tissue development over several days. However, high time-resolution time-series analysis makes it difficult to quantify phenotypes because it requires processing a large number of images. LRP development is classified into stages I–VII based on morphological features. Therefore, it requires the manual annotation of thousands of LRP time-lapse images of different stages, times and genotypes. We solved this problem by using DNNs to automatically quantify phenotypes, using Resnet50 as the DNN model that was developed by training approximately 8,000 images to create this model (Figure 1). This method captures the transition time between LRP stages. The assessment of LRP development using time-lapse image analysis showed that developmental delay occurs in the later stages of LRP development in *kcs1-5* and *MYB93* overexpressors (Figure 1). In this study, we also conducted LR induction

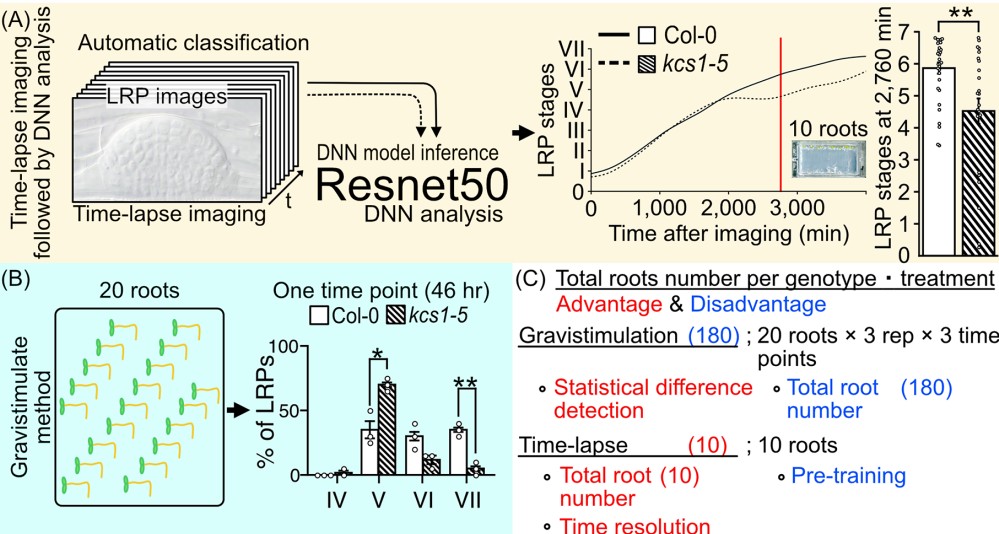

**Figure 1.** Comparison of conventional lateral root primordia (LRP) stage analysis methods (gravistimulation method) with a time-lapse imaging followed by a deep neural network (DNN) analysis method. (a) Schematic model of time-lapse imaging followed by the DNN analysis. LRP images obtained through time-lapse imaging serve as input for machine learning, facilitating the construction of a DNN model (in our case, utilising Resnet50). Subsequently, approximately 10 roots from experimental samples are subjected to time-lapse imaging to capture the developing LRP. By employing the established DNN model, automated identification of LRP stages is executed. The LRP stage transitions are then depicted graphically over time on the *x*-axis. Solid and dashed lines indicate LRP stage transition times for Col-0 and *kcs1-5*, respectively. The right bar graph presents statistical analysis results of DNN analysis at 2,760 min (46 h). The *y*-axis indicates the LRP stage determined using DNN. White box, Col-0; hatched box, *kcs1-5*. Significant difference from Col-0 was determined using the Welch's *t*-test (∗∗*P* = 0.004). The data were retrieved from Uemura et al. (2023). (b) Schematic model of the gravistimulation method. In each bio-replication experiment, at least 20 roots are subjected to gravity stimulation at a single time point. Following clearing of roots, the LRP stages are counted under a microscope. The right bar graph is the percentage of LRP at different stages of development after 46 h (2,760 min) of gravistimulation. Data are presented as mean ± SE of three biological replicates, with 20 seedlings in each replicate. White box, Col-0; hatched box, *kcs1-5*. Significant differences from Col-0 were determined using the generalised liner mixed model followed by Holm's *P*-value adjustment in each stage (∗∗*P* < 0.01, ∗*P* < 0.05). Similar to the bar graph in (a), the transition to the later stages of LRP development (stages VI and VII) was delayed in *kcs1-5*. The data were retrieved from Uemura et al. (2023). (c) Comparison of these two methods. One advantage of gravistimulation method is its high capacity to detect statistical differences. Conversely, a disadvantage is that as the number of time points, genotypes and treatments becomes enormous, data analysis becomes more complex. In the gravistimulation method, a minimal observation of at least 180 roots is required for a single genotype or treatment at a given time. The advantage of time-lapse DNN lies in conducting imaging with temporal information, requiring fewer individuals. With DNN, time-lapse imaging of 10 roots for a single genotype and treatment can be conducted, incorporating temporal information. However, a disadvantage is the need for pre-training the DNN with a substantial number of LRP images.

experiments via gravistimulation and compared the results using the DNN analysis. *kcs1-5* and *MYB93* overexpressors exhibited reduction in the number of LRP in the later stages, consistent with the DNN analysis results (Figure 1). In the gravistimulation assay, the number of LRP in at least 180 roots was measured under one condition, with biological replicates of three sets of 20 roots at three different time points for a single genotype or chemical treatment. Therefore, when considering mutants, overexpressors, and each chemical treatment, the total number exceeds 1,000 roots. Combined LRP time-lapse imaging and DNN phenotypic analysis quantitatively indicated the phenotypic changes over time from only 20 roots. Moreover, using our system, we could classify LRP stages in not only mutants but also plants subjected to any chemical treatment. Compared with the gravistimulation assay, our method offers major advantages in terms of reducing labour, improving stage discrimination accuracy and increasing research speed. However, to construct a DNN model, it is imperative to initially acquire a substantial number of images. In our case, after acquiring over 8,000 images of various LRP stages, we classified the images manually and subsequently employed machine learning techniques. Once the training phase is validated, the analysis of newly captured time-lapse images becomes feasible with considerably fewer images. Moreover, it is important to analyse and interpret continuous images captured via time-lapse imaging to elucidate seamless plant development. We have shown that it

is effective to quantify LRP development using a DNN with time-lapse imaging. Even in time-lapse imaging, this technique is useful for studying other tissues and cells. This is because it enables the observation of spatiotemporal changes such as molecular signalling, organelle behaviour and organ development along with the imaging of certain reporters. However, our analysis was conducted only in *Arabidopsis*. For other plant species with larger root tissues than *Arabidopsis*, optimisation of imaging and other preparations might be essential.

Time-lapse imaging is a tool for capturing a substantial number of images over time. The development and growth processes of plants undergo continuous changes, making it necessary to analyse morphological changes accurately by considering multiple time points. However, a gap persists owing to a lack of technique for the automatic quantification and statistical processing of phenotypic information. In plants, deep learning has been leveraged for tasks such as classification and segmentation. This facilitates the automated identification of disease species through the analysis of images depicting plant diseases and enumeration of leaf numbers based on shoot images (Singh et al., 2018; Ubbens & Stavness, 2017; Yuan et al., 2023). The application of deep learning has streamlined the quantification of phenotypic information from images, which was previously challenging. Through the quantitative analysis of extensive image datasets, including time-lapse LRP images, insights can be gleaned from the wealth of temporal information.

## Acknowledgements

We would like to thank Editage for English language editing.

**Funding statement.** This work was supported by the Ministry of Education, Culture, Sports, Science, and Technology (MEXT) KAKENHI (Grant No. JP23H04207).

**Competing interest.** The authors declare no competing interests.

**Author contributions.** This manuscript was conceived and written by Y.U. and H.T.

**Data availability statement.** No data or code were developed for this manuscript.

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
