## [Reviewer Report]

[20230728]

Editorial Team

Quantitative Plant Biology

Dear Editor:

I wish to submit an “insight” for publication in Quantitative Plant Biology titled “Quantitative analysis of lateral root development with time-lapse imaging and deep neural network.” The paper was coauthored by Yuta Uemura and Hironaka Tsukagoshi.

This insight paper aimed to discuss an effective technique to quantify the organ morphology using a combination of time-lapse imaging and deep neural network analysis. We believe that this paper will be of interest to the readership of your journal because time-lapse imaging can help elucidate the processes involved in plant development, and integrating it with DNN offers an effective technique to quantify the timing of the transition of organ morphology. Moreover, the technique we developed can help elucidate spatiotemporal molecular mechanisms in plant development.

All study participants provided informed consent, and the study design was approved by the appropriate ethics review board. We have read and understood your journal’s policies, and we believe that neither the manuscript nor the study violates any of these. There are no conflicts of interest to declare.

Thank you for your consideration. I look forward to hearing from you.

Sincerely,

Hironaka Tsukagoshi

Faculty of Agriculture

Meijo University

1-501 Shiogamaguchi, Tempaku-ku, Nagoya 

Aichi, 468-8502 Japan

thiro@meijo-u.ac.jp

---

## [Reviewer Report]

In the current insight, Uemura and Tsukagoshi make a case for quantitative analysis and scoring of lateral root development (in Arabidopsis). The authors make a case for the use of Deep Neural Networks (DNN) to score LRs in an automatic and non-supervised manner. They mostly refer to their recently published study (Uemura et al., 2023 TPJ) to make a case for this approach and propose that the use of DNN should be more explored for quantitative analysis of LR development (in this case) or any other developmental process. I think this insight would benefit from a figure that shows the different approaches one can use to study a given developmental process (confocal, cleared tissue, time lapse etc etc) and what the benefits/limitations are of each approach. This could give a clear insight as to what the power of the use of DNN is, but also for which traits it might not be so straightforward to implement a similar approach. I think this would make the insight a little bit more balanced and also more interesting for the general reader.

---

## [Reviewer Report]

This manuscript is an “insight paper” – my understanding is that this is a short review of a key paper with additional insights and descriptions of how it connects to broader literature. This particular insight describes the work in Uemura et al, which used deep neural network (DNN) analysis to increase the speed of analysis of lateral root development in plants. The authors used their DNN pipeline to compare lateral root development in wildtype and mutants in the very-long chain fatty acid (VLCFA) pathway. The insight paper is interesting because it’s focus is very different from the original manuscript – the focus of the insight paper is on the DNN, but the original paper focuses much more on the biology. I think this is overall a strong approach, but there are some important ways that this paper could be improved.

1) It is not clear from reading this paper what the intention is. There is a brief mention in the abstract, but if you imagine someone downloading this paper off the internet without being familiar with “insight papers,” I think they might be very confused. To make the intention clearer, I think the introduction should include a description of the objective(s) of this manuscript. I also think that the main paper that is the focus of the review should be highlighted, instead of being cited like a normal paper. This will help it stand out to the reader.

2) Although I appreciate that the objective of this paper seems to be to focus on DNNs as a valuable research approach, there is a lot of information in the introduction about the biology of lateral root development. However, the biology discovered using a DNN is not really described (for instance, I had to read the source paper to learn that VLCFA is a regulator of LRP development through transcription factor-mediated regulation of gene expression and the transportation of VLCFAs is also involved in LR development through root cap cuticle formation). It’s a little bit of a disconnect. My recommendation is to match the introduction of the biology with a description of the results: so either shorten the biology descriptions in the introduction and focus instead on DNNs, or to have a longer summary of the biological results towards the middle/end of the manuscript.

3) There is a large gap between when DNN is introduced and lines 121 – 148, which describes why DNN was chosen for this type of analysis. I think the flow of the manuscript might be better if you raise this discussion earlier (maybe right after line 103, where you first introduce DNN).

4) Given that the focus of this manuscript is DNN analysis, I think it’s worth describing how the DNN performed compared to current gold standard methodology (how it was validated). This information is in the original paper, but a summary here would be useful. In addition, the source code could also be provided, for interested readers.

Minor comments:

There is need for editing for grammar and clarity throughout the document. Here are some important examples:

Line 59: recommend rewording to say “de novo tissue development” as opposed to “novel tissue development.”

Line 60: I’m not sure what the authors mean by “the lateral root temporally transitions between cellular and tissue states.” Perhaps adding citations would clarify this statement.

Line 64: I think that the authors are alluding to the fact that most labs study root development at a single time point, but especially for lateral roots, this is a dynamic process where time is an important component. This could be more clearly explained.

Line 97: The gravistimulated roots assay description should include a reference.

Line 101: This statement (“The combination of time-lapse imaging and …”) is a really critical part of the manuscript. This is where I think your central paper should be highlighted. The tense here is a bit strange – I think it might be clearer to say something along the lines of:

“We reasoned that the combination of time-lapse imaging and DNN analysis would make it possible to measure the developmental transitions of a single LRP. Therefore, we developed a machine learning-based method to quantify LRP development, as published in ‘A very long chain fatty acid responsive transcription factor, MYB93, regulates lateral root development in Arabidopsis’ (Uemura et al., The Plant Journal, 2023). Here, we describe the rational, development, and implications of the DNN approach for lateral root studies as well as explorations of many other biological questions.”

---

## [Reviewer Report]

Dear Dr. Tsukagoshi,

thank you for your submission to QPB. The reviewers found your study interesting, however, they found several issues that need to be improved. I hope that you find the comments useful. 

Looking forward to read the revised version of your manuscript in the near future.

with best wishes,

Ari Pekka Mähönen

---

## [Reviewer Report]

20231129

Editor-in-Chief 

Dr. Olivier Hamant

Quantitative Plant Biology 

Dear Editor: 

We/I wish to re-submit the manuscript titled “Quantitative analysis of lateral root development with time-lapse imaging and deep neural network.” The manuscript ID is QPB-23 -0007.

We thank you and the reviewers for your thoughtful suggestions and insights. The manuscript has benefited from these insightful suggestions. I look forward to working with you and the reviewers to move this manuscript closer to publication in the Quantitative Plant Biology journal.

The manuscript has been rechecked and the necessary changes have been made in accordance with the reviewers’ suggestions. The responses to all comments have been prepared and attached herewith. 

Thank you for your consideration. I look forward to hearing from you.

Sincerely,

Hironaka Tsukagoshi

Faculty of Agriculture, Meijo University, Nagoya, Aichi, Japan

E-mail: thiro@meijo-u.ac.jp

---

## [Reviewer Report]

Overall, these review address my main concerns. I have one more suggestion:

Line 55: I’m still not exactly sure what the authors mean by “the LRP temporally transitions through cellular states and finally forms lateral root (Torres-Martínez et al., 2019).” I think this means that in comparison to the mature primary root, which maintains a distinct and stereotyped set of cell lineages in the meristem, the cells in the developing lateral root are dynamically undergoing changes in identity, at least until the root matures.

---

## [Reviewer Report]

It serves its purpose as an insight. It would have been nice to have included a figure showing results obtained with the classical, gravistimulation method, and their DNN approach coupled to the live imaging. This could help the reader get insights into the results obtained when phenotyping the same genotypes with two different methods. In addition, it might be good to mention that the described approaches are mostly suited for Arabidopsis and perhaps not for species with larger roots or where gravistimulation might not work due to a more complex vascular system.

---

## [Reviewer Report]

Dear Prof. Tsukagoshi,

the two reviewers have now evaluated your manuscript, and overall the manuscript is getting close to acceptance. However, I agree with a reviewer that there is still a small textual issue (see details in reviewer’s comment) that needs to be handled before the acceptance. 

best wishes,

Ari Pekka Mähönen

---

## [Reviewer Report]

[20240116]

Olivier Hamant

Editor-in-Chief 

*Quantitative Plant Biology*

Ari Pekka Mahönen

Associate Editor

*Quantitative Plant Biology*

Dear Editors: 

We wish to re-submit the manuscript titled “Quantitative analysis of lateral root development with time-lapse imaging and a deep neural network analysis.” The manuscript ID is QPB-23-0007.R1.

We thank you and the reviewers for your thoughtful suggestions and insights. The manuscript has benefited from these insightful suggestions. We look forward to working with you and the reviewers to move this manuscript closer to publication in *Quantitative Plant Biology*.

The manuscript has been rechecked and the necessary changes have been made in accordance with the reviewers’ suggestions. The responses to all comments have been prepared and submitted along with the revised manuscript. 

Thank you for your consideration. We look forward to hearing from you.

Sincerely,

Hironaka Tsukagoshi

Faculty of Agriculture, Meijo University, Nagoya, Aichi, Japan 

Phone: +81-52-838-2372

E-mail: thiro@meijo-u.ac.jp